# The Secretome of Human Dental Pulp Stem Cells and Its Components GDF15 and HB-EGF Protect Amyotrophic Lateral Sclerosis Motoneurons against Death

**DOI:** 10.3390/biomedicines11082152

**Published:** 2023-07-30

**Authors:** Richard Younes, Youssef Issa, Nadia Jdaa, Batoul Chouaib, Véronique Brugioti, Désiré Challuau, Cédric Raoul, Frédérique Scamps, Frédéric Cuisinier, Cécile Hilaire

**Affiliations:** 1INM, University of Montpellier, INSERM, 34295 Montpellier, France; richard.younes@inserm.fr (R.Y.); youssef.issa@inserm.fr (Y.I.); nadia.jdaa@hotmail.fr (N.J.); veronique.brugioti@inserm.fr (V.B.); desire.challuau@inserm.fr (D.C.); cedric.raoul@inserm.fr (C.R.); frederique.scamps@inserm.fr (F.S.); 2LBN, University of Montpellier, 34193 Montpellier, France; batoul_chouaib@hotmail.com (B.C.); frederic.cuisinier@umontpellier.fr (F.C.); 3Neuroscience Research Center, Faculty of Medical Sciences, Lebanese University, Beirut 6573, Lebanon; 4Human Health Department, IRSN, SERAMED, LRMed, 92262 Fontenay-aux-Roses, France

**Keywords:** amyotrophic lateral sclerosis, neuropathology, death, axon outgrowth, electrical activity, dental pulp stem cell, conditioned medium, neurotrophic factors, motoneuron, cell therapy

## Abstract

Amyotrophic lateral sclerosis (ALS) is a fatal and incurable paralytic disorder caused by the progressive death of upper and lower motoneurons. Although numerous strategies have been developed to slow disease progression and improve life quality, to date only a few therapeutic treatments are available with still unsatisfactory therapeutic benefits. The secretome of dental pulp stem cells (DPSCs) contains numerous neurotrophic factors that could promote motoneuron survival. Accordingly, DPSCs confer neuroprotective benefits to the *SOD1^G93A^* mouse model of ALS. However, the mode of action of DPSC secretome on motoneurons remains largely unknown. Here, we used conditioned medium of human DPSCs (DPSCs-CM) and assessed its effect on survival, axonal length, and electrical activity of cultured wildtype and *SOD1^G93A^* motoneurons. To further understand the role of individual factors secreted by DPSCs and to circumvent the secretome variability bias, we focused on GDF15 and HB-EGF whose neuroprotective properties remain elusive in the ALS pathogenic context. DPSCs-CM rescues motoneurons from trophic factor deprivation-induced death, promotes axon outgrowth of wildtype but not *SOD1^G93A^* mutant motoneurons, and has no impact on the spontaneous electrical activity of wildtype or mutant motoneurons. Both GDF15 and HB-EGF protect *SOD1^G93A^* motoneurons against nitric oxide-induced death, but not against death induced by trophic factor deprivation. GDF15 and HB-EGF receptors were found to be expressed in the spinal cord, with a two-fold increase in expression for the GDF15 low-affinity receptor in *SOD1^G93A^* mice. Therefore, the secretome of DPSCs appears as a new potential therapeutic candidate for ALS.

## 1. Introduction

Amyotrophic lateral sclerosis (ALS) is an incurable neurodegenerative disease that leads to the selective loss of upper and lower motoneurons resulting in progressive paralysis and death from respiratory failure usually within 3 years of diagnosis. It remains one of the most devastating neurodegenerative diseases to date, due to the lack of effective treatment.

Approximately 90% of cases are sporadic, with no known family history of the disease. Sporadic and inherited forms, referred to as familial, are mostly indistinguishable by clinical and pathological markers. ALS is primarily characterized by degeneration of motoneurons in the brain and spinal cord [1]. Numerous studies have established that cellular and molecular pathophysiological mechanisms act concomitantly or sequentially to lead to neuronal dysfunction and, ultimately, death [2]. These include abnormal aggregates of misfolded proteins, defects in axonal transport and RNA metabolism, mitochondrial dysfunction, oxidative stress, glutamate excitotoxicity, and neuroinflammation [3,4,5,6,7]. These different molecular pathways, as well as the different cell types, represent different therapeutic targets, thus underlining the complexity of developing effective therapeutic strategies.

More recently, echoing clinical findings, neuroimaging and neuropathological studies in humans have revealed a decrease in the volume of several subcortical gray nuclei, including the thalamus, hippocampus, amygdala, caudate nucleus, and nucleus accumbens [8]. Cerebellar dysfunction has also been reported [9]. The thalamus is affected, not only in its motor part but also in areas associated with learning and memory encoding, emotional, cognitive, and sensory processes [10]. As a result, this disease is now often regarded as part of the frontotemporal dementia spectrum, which is in line with the clinical picture, since a significant proportion of patients exhibit, in addition to motor pyramidal symptoms, extrapyramidal, cognitive, and sensory symptoms [10]. For the moment, the cellular and molecular mechanisms affecting these structures in ALS are not yet known. Studies in ALS mouse models also show that at presymptomatic stages of the disease, changes in the motor thalamus are already present, with neuronal death in motor nuclei and mild gliosis [11]. A cerebellar pathology is also present [9].

The identification of new areas and specific brain circuits affected by ALS can eventually help us to develop new, more complex preclinical models that better reflect the heterogeneity of the disease and thus improve clinical transfer, to develop new treatments [12,13]. Meanwhile, it may also help us to continue dissecting the various pathophysiological mechanisms leading to motoneuron death in known mouse models of the disease, and to decipher new molecular pathways to discover other potential therapeutic targets.

Despite our better pathophysiological identification of the disease, to date only two approved drugs, riluzole and edaravone, are used to treat ALS patients [7]. Riluzole has mainly an anti-glutamatergic effect [14,15] and edaravone is a free-radical scavenger that protects against oxidative stress [16,17,18]. Unfortunately, the clinical outcomes of these two treatments are still unsatisfactory for patients. These two approved drugs only modestly slow disease progression and extend the patient’s life by few months. A phase III clinical trial has evaluated antisense oligonucleotides (ASOs) targeting SOD1. The administration of ASOs in patients led to a decrease in SOD1 levels in the cerebrospinal fluid and the neurofilament light chain biomarker in the plasma [19]. However, ASO delivery did not significantly change the revised ALS functional rating scale score, the main clinical endpoint.

Another strategy for developing new therapies is to focus on neurotrophic factors (NTFs). Indeed, NTFs are essential for the maintenance, survival, neurite outgrowth, and axonal regeneration of motoneurons during development and adulthood [20,21]. The pleiotropic characteristics of NTFs have made them attractive therapeutic candidates for ALS to restore neuromuscular synapses and promote motoneuron survival. Unfortunately, after thirty years of clinical trials based on the administration of recombinant NTFs, the clinical outcomes remain largely disappointing [22]. The choice of route of administration, passage through the blood–brain barrier, and the limited half-life of NTFs in the blood are parameters that obviously need to be considered for these recombinant proteins. The NTFs to be evaluated, which when delivered to the periphery can lead to undesirable effects, also raise questions about methods for targeting the central nervous system, such as that based on viral gene transfer [23]. Another consideration is that of using a combinatorial approach to NTFs to optimize therapeutic effects by targeting different cell populations. Therapies based on complex protein solutions remain interesting to explore, either directly as biomaterial mixtures of therapeutic factors or as a means of identifying new pro-survival factors.

Dental pulp stem cells (DPSCs) secrete numerous NTFs that give them potent neuroprotective properties [24,25,26]. Moreover, several studies have shown the potential of these cells or their secretome in the treatment of neurodegenerative diseases [27,28,29,30].

Growing evidence has demonstrated the potential of DPSCs for the treatment of various diseases and dysfunctions, including neurodegenerative diseases. DPSCs can be used either directly as differentiated neurons or via their secretome. A major interest in the use of DPSCs lies in their isolation from the human third molar (wisdom tooth), since human molars are easy to extract and are considered medical waste. DPSCs proliferate rapidly and have the ability to differentiate into various cell types. Moreover, their secretome contains growth and neurotrophic factors. This secretome or conditioned medium (CM) can replace cells which, due to their ability to multiply, in some cases can be a source of cancer [31,32]. The potential therapeutic value of DPSCs has been demonstrated in spinal cord injury [33], traumatic brain injury [34], stroke [35], cerebral ischemia [36], Alzheimer’s disease [30], Parkinson’s disease [27], aneurysmal subarachnoid hemorrhage [37], retinal lesions [38], and ALS [29].

For ALS, the pioneering study by Wang et al. showed that intraperitoneal injection of DPSC secretome in an ALS mouse model expressing the ALS-causing mutation G93A in superoxide dismutase 1 (SOD1^G93A^) increased lifespan and the number of surviving motoneurons [29]. However, the mechanism of action of DPSC secretome on motoneurons remains largely unknown. Moreover, DPSC secretome is a complex medium containing many factors [39]. Its composition is difficult to control and largely depends on manufacturing methods and protocols [40]. Identifying individual components is therefore an important approach to reducing constraints on the use of this complex media.

In this study, we first studied the effects of DPSC secretome on the survival, axon outgrowth, and synaptically driven electrical activity of mouse primary motoneurons from wildtype and *SOD1^G93A^* mice. We then evaluated the neuroprotective potential of two candidate molecules that we selected from our previous work [39], growth differentiation factor 15 (GDF15) and heparin-binding EGF-like growth factor (HB-EGF).

We show that the secretome of DPSCs promotes the survival of wildtype and *SOD1^G93A^* motoneurons without altering their electrical activity. DPSC secretome also enhances axon outgrowth of wildtype but not *SOD1^G93A^* motoneurons. Interestingly, both GDF15 and HB-EGF confer neuroprotection to SOD1^G93A^-expressing motoneurons against nitric oxide (NO)-induced cytotoxicity. Altogether, our results propose new therapeutic perspectives to explore based on DPSC secretome content, GDF15, and HB-EGF.

## 2. Materials and Methods

### 2.1. Preparation of Conditioned Media

#### 2.1.1. DPSC-Conditioned Media

DPSCs were obtained from the wisdom teeth of a 15-year-old patient. Informed consent was obtained from the patient after receiving approval by the local ethics committee. DPSCs were recovered as previously described [41,42]. Briefly, after being disinfected, the teeth were cut along the cementum–enamel junction using a diamond disc and were split in two parts. Pulps were then collected and incubated with 3 mg/mL of type I collagenase and 4 mg/mL dispase for 1 h. Digested pulps were filtered, centrifuged, and the pellet was resuspended and cultured in α-MEM (ThermoFisher Scientific, Waltham, MA, USA) with 1% penicillin–streptomycin, 10% fetal bovine serum, and 1 µg/mL of recombinant human basic fibroblast growth factor (R&D System, Minneapolis, MN, USA). The culture medium was changed after 24 h and then changed twice a week [39]. DPSCs were allowed to multiply until they reached 80–90% confluency and then split with 0.05% trypsin-EDTA (ThermoFisher Scientific, Waltham, MA, USA) for 3 min at 37 °C to enhance the colony. After the 3rd or 4th passage, when DPSCs reached 90% confluency, cells were washed twice with phosphate buffered saline (PBS) and culture medium was replaced with serum-free Neurobasal medium containing 50 µM L-glutamine, 2% B-27 supplement, and 1% penicillin–streptomycin (ThermoFisher Scientific, Waltham, MA, USA). After 48 h of culture, the medium was collected, centrifuged once at 450× *g* for 5 min and then centrifuged at 1800× *g* for 3 min to remove cell debris. Unless used fresh, the conditioned medium was stored at −80 °C [39].

#### 2.1.2. Adipose Derived Stems Cells (ASCs) and Fibroblast-Conditioned Media

Human adipose tissue was provided by the Institute for Regenerative Medicine and Biotherapy (IRMB, Montpellier, France) and human skin fibrobasts were given by Dr. Vasiliki Kalatzis (INM, Inserm UMR1298, Montpellier, France). The conditioned medium for these two cell types was produced in the same way as for DPSCs and used at passage 4 for ASCs and passage 6 for fibroblasts.

### 2.2. Animals

All animal experiments were approved by the national ethics committee on animal experimentation, and were conducted in compliance with the European Community and national directives for the care and use of laboratory animals. B6.Cg-Tg(SOD1*G93A)1Gur/J (*SOD1^G93A^*) mice and B6.Cg-Tg(Hlxb9-GFP)1Tmj/J (*Hb9::GFP*) mice were purchased from the Jackson laboratory and maintained on a C57BL/6J background under specific-pathogen-free conditions. They were housed in cages with a 12 h light/12 h dark cycle with food and water supplied ad libitum in our animal facility accredited by the French Ministry of Food, Agriculture and Fisheries (B-34 172 36-11 March 2010). Experiments were conducted in accordance with the Directives of the Council of the European Communities of November 24, 1986 (86/609/EEC) and the French Ethics Committee (approval A34-506). Thirty-seven wildtype mice and twenty-three SOD1^G93A^ mice were used for embryonic motoneuron primary cultures. Five wildtype and five SOD1^G93A^ 3-month-old mice were used for quantitative RT-PCR.

### 2.3. Motoneurons Immunopurification and Culture

Motoneurons were purified from the spinal cords of wildtype and *SOD1^G93A^* embryos using 5.2% iodixanol density gradient centrifugation combined with anti-p75-based magnetic cell isolation (Miltenyi Biotec, Bergisch Gladbach, Germany, clone REA648) as we previously described [43,44]. Motoneurons were plated on poly-ornithine (3 µg/mL)/laminin (2 µg/mL)-coated 4-well plates in supplemented Neurobasal medium containing 2% (*v*/*v*) horse serum, 25 µM L-glutamate, 25 µM β-mercaptoethanol, 50 µM mM L-glutamine, 2% (vol/vol) B-27 supplement, and 0.5% penicillin–streptomycin (ThermoFisher Scientific, Waltham, MA, USA). When indicated, a cocktail of neurotrophic factors (0.1 ng/mL GDNF (Sigma-Aldrich, Saint-Louis, MO, USA, G1401), 1 ng/mL brain-derived neurotrophic factor (BDNF) (ImmunoTools, MGC34632), and 10 ng/mL CNTF (R&D Systems, Minneapolis, MN, USA, 557-NT/CF)) was added to the supplemented Neurobasal medium. Recombinant mouse GDF15 (R&D Systems, Minneapolis, MN, USA, 8944-GD) and HB-EGF (E4643, Sigma-Aldrich, Saint-Louis, MO, USA) were added at the time of seeding in basal supplemented Neurobasal medium. DETANONOate (Enzo Life Sciences, Farmingdale, NY, USA, ALX-430-014) was added after 24 h of culture.

For the electrophysiological recordings, motoneurons were isolated from the ventral spinal cord of *Hb9::GFP* and *SOD1^G93A^* E12.5 embryos using iodixanol density gradient centrifugation [45].

### 2.4. Motoneurons Survival

Motoneurons were seeded at a density of 1250 cells/cm^2^. Wildtype and *SOD1^G93A^* immunopurified motoneurons were counted using a phase contrast microscope using morphological criteria. Motoneurons are considered as living cells if their axons are more than three times the length of the cell body, if those axons are not completely degraded, and if the cell body does not contain vacuoles [44,46]. Counting was performed on three or four separate samples, and the number of surviving motoneurons was determined after 24 h of culture. To allow comparison of values from different experiments, survival values were normalized relative to the value in the absence of neurotrophic factors.

### 2.5. Axon Length Measurements

For the quantification of axon length, motoneurons were seeded at a density of 750 cells/cm^2^. At one day in vitro, motoneurons were processed for immunostaining with β-tubulin III antibodies. Images were acquired with a ZEISS AXIO Imager Z2 Apotome and axon length was determined using ImageJ software v1.53 and the NeuronJ plugin (National Institutes of Health, Bethesda, MD, USA) [47]. Total axon length was determined by measuring the length of the longest neurite with connected branches of β-tubulin III-positive motoneurons [46].

### 2.6. Electrophysiology

For electrophysiological analysis, motoneurons were seeded at a density of 1250 cells/cm^2^. Spontaneous electrical activity was recorded at room temperature using the loose-patch electrophysiological technique with an Axopatch 200B amplifier (Molecular Devices, San José, CA, USA). The bathing solution contained 145 mM NaCl, 5 mM KCl, 10 mM D-glucose, 10 mM HEPES, 2 mM CaCl2, and 1.5 mM MgCl2, adjusted to pH 7.4 and 310 mosm. The electrode was filled with the same extracellular solution and to obtain a good resolution of extracellular recordings of the spontaneous activity, the contact with the cell membrane had a resistance in the range of 30–100 MΩ. Cultures were maintained at 37 °C, 7.5% CO_2_. All recordings were conducted between 7 and 9 days of culture [45,48]. Motoneurons were identified according to morphological criteria (large size, multipolar with high dendritic complexity) or GFP expression [44].

### 2.7. Immunocytochemistry

Motoneurons were first fixed with 4% paraformaldehyde (PFA) in PBS that was directly added to the culture medium (1:1) for 10 min, and then fixed with 4% PFA in PBS for 15 min on ice. Cells were then washed with PBS and incubated for 1 h in blocking solution containing 4% bovine serum albumin, 4% donkey serum, and 0.1% Triton-X100 in PBS. Coverslips were then incubated overnight at +4 °C with rabbit anti-βIII tubulin (Sigma-Aldrich, Saint-Louis, MO, USA, T2200). Cells were washed 3 times for 10 min each with PBS, incubated with the appropriate fluorescent-conjugated secondary antibody (ThermoFisher Scientific, Waltham, MA, USA), washed, and mounted in Moviol solution [45]. Image acquisition was carried out on a Zeiss (Car Zeiss AG, Oberkochen, Germany) Axio Imager Z2-module Apotome 2.0. ImageJ (National Institutes of Health, Bethesda, MD, USA) was used for axon tracing.

### 2.8. Immunohistochemistry

As previously described [43] for the spinal cord, mice were anaesthetized and transcardially perfused with 4% PFA in PBS. The lumbar spinal cord was removed, post-fixed in 4% PFA for 2 h, incubated in 30% sucrose in PBS, embedded in OCT, flash-frozen, and cut at 18 µm thickness. The sections were then rinsed for 5 min with PBS and incubated for 2 h at room temperature in blocking solution (PBS, 5% donkey serum, 0.3% Triton-X100, 0.05% Tween-20). This was followed by overnight incubation at +4 °C with the following primary antibodies: rabbit anti-TGFβ-R1 (Sigma-Aldrich, Saint-Louis, MO, USA, SAB4502958, 1:200) and goat anti-choline acetyl transferase (ChAT) (Merck, Darmstadt, Germany, AB144P, 1:100). Sections were washed and incubated for 1 h at room temperature with appropriate AlexaFluor-conjugated secondary antibodies (ThermoFisher Scientifc, Waltham, MA, USA). Slides were mounted in Mowiol solution. Image acquisition was carried out on a Zeiss Axio imager Z2-module Apotome 2.0 (Car Zeiss AG, Oberkochen, Germany).

### 2.9. Quantitative Reverse Transcription Polymerase Chain Reaction

For the lumbar spinal cord, 90-day-old mice were deeply anaesthetized, perfused transcardially with PBS, and tissue was harvested in RNAprotect tissue reagent (Qiagen, Hilden, Germany). Lysis buffer was used to homogenize the tissue that was then passed through needles. The lysates were mixed with an equal volume of 70% ethanol, and total mRNA was separated from other cellular components on RNeasy minispin columns. The eluted mRNA was quantified spectrophotometrically (Nanodrop). After removal of genomic DNA, reverse transcription was performed with 1 µg of mRNA using the Quantitect RT kit (Qiagen, Hilden, Germany). The collected cDNA was diluted to 100 ng in H_2_0 and stored at −20 °C until use. Quantitative PCR was performed on 10 ng of cDNA with SYBR green dye (Qiagen, Hilden, Germany) using the LightCycler 480 system (Roche Diagnostics, Basel, Switzerland). The primers used were as follows: *Egfr*, 5′-ATTAATCCCGGAGAGCCAGA-3′ and 5′-TGTGCCTTGGCAGACTTTCT-3′; *ErbB4*, 5′-TCCACTTTACCACAACACGCT-3′ and 5′-TCAAAGCCATGATCACCAGGA-3′; *Tgfbr1,* 5′-ATGTGGAAATGGAAGCCCAGA-3′ and 5′-ATGACAGTGCGGTTATGGCA-3′; *Tgfbr2* variant 1, 5′-TGTTGAGATTGCAGGATCTGG-3′ and 5′-TGGACAGTCTCACATCGCAA-3′; *Tgfbr2* variant 2, 5′-TTCCCAAGTCGGTTAACAGTGA-3′ and 5′-TTCTGGTTGTCGCAAGTGGA-3′; *Polymerase (RNA) II polypeptide J* (*Polr2j*), 5′-ACCACACTCTGGGGAACATC-3′ and 5′-CTCGCTGATGAGGTCTGTGA-3′. The PCR conditions were 15 s at 94 °C, 20 s at 60 °C, 35 s at 72 °C for a total of 45 cycles. After PCR amplification, melting curve analysis was performed to verify PCR specificity. The level of the domestic gene *polymerase (RNA) II polypeptide J* (*Polr2J*) was used to normalize cDNA amounts. Ct was calculated as the difference between Ct values, determined with Equation (2)-Ct [48].

### 2.10. Statistical Analysis

Data are presented as means ± standard error of the mean (SEM) (Appendix A). Statistical analysis was performed using Prism 8 (GraphPad, San Diego, CA, USA). One-way ANOVA was used for multiple comparisons followed by Tukey’s *post hoc* test. For qRT-PCR experiments, the Mann–Whitney test was used. Statistical significance was accepted at the level of *p* < 0.05 (Appendix A).

## 3. Results

### 3.1. DPSCs-CM Promotes the Survival of Motoneurons

We first aimed to determine whether DPSCs-CM can have an effect on motoneuron survival under deprivation of NTFs. For this purpose, we studied the effect of increasing concentrations of DPSCs-CM in the culture medium without the standard cocktail of GNDF, BDNF, and CNTF. These NTFs are known to promote optimal survival of embryonic motoneurons [44]. The efficacy of the different concentrations of DPSCs-CM was assessed 24 h after seeding and motoneuron survival was expressed relative to the control condition where motoneurons were cultured in the absence of NTFs. We did not observe any neurotrophic effect for 5 and 10% DPSCS-CM concentrations (Figure 1A). However, the survival rate was 2 to 2.5 times higher when motoneurons were cultured in the presence of 25, 50, and 75% DPSCs-CM. We did not find any significant difference in motoneuron survival between the conditions with 25, 50, and 75% concentrations or with the NTFs. However, when motoneurons were cultured only in the presence of DPSCs-CM (100%), a detrimental effect on survival was observed (Figure 1A). Our results indicate that DPSCs-CM rescue motoneurons from death induced by trophic deprivation.

The DPSCs are neural crest-derived mesenchymal stem cells originating from the ectoderm [49]. To determine whether the neurotrophic effect we observed was specific to DPSCs-CM, we analyzed the neurotrophic properties of CMs derived from human ASCs (ASCs-CM), which are mesenchymal stem cells originating from the mesoderm, and human fibroblasts (Fibro-CM). Following 24 h of culture, we did not observe any neurotrophic effect of ASCs-CM at 50 or 100% on motoneuron survival compared with the NTF-free medium (Figure 1B). Similarly, 50 and 100% Fibro-CM did not improve the survival of motoneurons compared with the NTF-free medium (Figure 1C). These results demonstrate that DPSCs-CM has selective neurotrophic properties for embryonic motoneurons.

We next investigated whether DPSCs-CM could also improve the survival of motoneurons purified from the SOD1^G93A^ ALS mouse model. The survival of SOD1^G93A^ motoneurons in the absence of NTFs after 24 h of culture was similar to that of wildtype motoneurons (not shown), as already described [44]. We then analyzed the survival of SOD1^G93A^ motoneurons in the optimal concentration of DPSCs-CM that we previously determined (Figure 1A). In the presence of 50% DPSCs-CM, we observed a significant increase in the percentage of surviving ALS motoneurons compared with the negative control (Figure 1D). There were no significant differences between the survival of motoneurons cultured in the presence of 50% DPSCs-CM and those cultured in the presence of NTFs.

Therefore, our results indicate that DPSCs-CM can provide a robust neurotrophic support to motoneurons expressing ALS-causing SOD1 mutation.

### 3.2. DPSCs-CM Promotes Axon Outgrowth of Wildtype, but Not SOD1^G93A^ Motoneurons

We next investigated whether DPSCs-CM could also influence neurite growth of wildtype and *SOD1^G93A^* motoneurons. Motoneurons were immunopurified from wildtype and *SOD1^G93A^* mouse embryos and immunostained with anti-βIII tubulin to trace the total length of the axon (including branchings) using the NeuronJ ImageJ plugin, as we previously described [47] (Figure 2A). We found that DPSCs-CM significantly increased the axon length of wildtype motoneurons cultured in the absence of NTFs (Figure 2B). Addition of DPSCs-CM to motoneuron culture elicited axon outgrowth as efficiently as the addition of the cocktail of NTFs. Interestingly, we observed that neither DPSCs-CM nor the cocktail of NTFs increased the total axon length of motoneurons (Figure 2C).

Our results highlight that an uncoupling between axonal outgrowth and survival promoted by the neurotrophic support of DPSCs-CM or recombinant NTFs occurs when an ALS causal gene is expressed in motoneurons.

### 3.3. DPSCs-CM Does Not Modify the Synaptically Driven Activity of Wildtype and SOD1^G93A^ Motoneurons

To complete the range of functional properties of DPSCs-CM on motoneurons, we then focused on the spontaneous electrical activity that results from the synaptic network activity. We performed extracellular recordings of the spontaneous firing rate of both wildtype and *SOD1^G93A^* neurons using the loose-patch technique. As we have previously shown, after 7 days in vitro (DIV), primary motoneurons exhibit adult-like intrinsic electrical activity and are spontaneously active due to a synaptic excitatory network [45,48] (Figure 3A). We did not observe any difference in the frequency of the spontaneous electrical activity of wildtype and *SOD1^G93A^* neurons cultured in the presence or the absence of NTFs (Figure 3B,C). When we then cultured neurons in the presence of DPSCs-CM (50%), we did not find any significant change in the spontaneous spike frequency (approximately 2 Hz in average) of either wildtype or SOD1^G93A^-expressing neurons (Figure 3B,C). Therefore, DPSC-CM does not modify the synaptically driven electrical activity of wildtype or *SOD1^G93A^* neurons.

### 3.4. GDF15 and HB-EGF Do Not Provide Any Neutrotrophic Support to Motoneurons

Among the wide palette of secreted proteins that we identified previously [39] we sought to focus on two proteins, GDF15 and HB-EGF, that have been described for their activity on neuronal survival, but whose trophic benefits for ALS motoneurons have not been evaluated [50,51]. Using a human growth factor antibody array we previously found that GDF15 and HB-EGF concentration amounted to 12 ± 12.8 pg/mL and 2 ± 4 pg/mL, respectively, in DPSCs-CM [39].

We first evaluated whether addition of recombinant GDF15 could promote the survival of wildtype motoneurons when cultured in the absence of NTFs. Motoneurons were treated with increasing concentrations of GDF15, based on what was previously described [51], and the percentage of surviving motoneurons determined after 24 h of culture. We found that GDF15 does not affect the survival of wildtype motoneurons (Figure 4A). When HB-EGF was evaluated for its prosurvival properties in motoneurons, using the optimal concentration previously described [50], we did not observe any neuroprotective benefits against NTF deprivation (Figure 4B).

We then asked whether GDF15 or HB-EGF might promote the survival of *SOD1^G93A^* motoneurons placed in basal conditions. We observed that neither GDF15 (10 ng/mL) nor HB-EGF (20 ng/mL) saved motoneurons from death induced by the absence of NTFs (Figure 4C).

From our data, neither GDF15 nor HB-EGF are able to rescue motoneurons from death induced by neurotrophic factor deprivation.

### 3.5. GDF15 and HB-EGF Prevent Motoneuron Death from Oxidative Insult

We also investigated whether these two candidates could rescue motoneuron death from oxidative stress. It was shown that motoneurons expressing SOD1 mutants have an exacerbated susceptibility to NO exposure [44]. Motoneurons purified from wildtype and SOD1^G93A^ mice were then cultured in the presence of NTFs, with GDF15 or HB-EGF, and exposed to the NO donor DETANONOate for 48 h. As previously described, DETANONOate induces death of mutant but not wildtype motoneurons (Figure 5A,B). Interestingly, we found that both GDF15 and HB-EGF rescued ALS motoneuron from death induced by NO (Figure 5B).

These data reveal a clear therapeutic potential of GDF15 and HB-EGF by rescuing motoneurons from death under pathological conditions.

### 3.6. Expression of GDF15 and HB-EGF in Adult Spinal Cord

To uncover the potential implication of HB-EGF and GDF15 signaling in ALS pathogenesis, we first analyzed the mRNA expression levels of their cognate receptors in the spinal cords of 3-month-old wildtype and SOD1^G93A^ mice using quantitative RT-PCR. At this presymptomatic stage, there is no substantial loss of motoneurons, but a stress response already takes place in the vulnerable population of motoneurons [52]. The transcript levels of the two HB-EGF receptors, Erbb4 and Egfr, were detected in the spinal cords of wildtype mice. The levels of *Egfr* remained unchanged in presymptomatic SOD1^G93A^ mice, while there was a significant two-fold decrease in *Erbb4* expression levels (Figure 6A). We found that the cognate high-affinity receptor of GDF15, Gfral, was not expressed in either the wildtype or SOD1^G93A^ spinal cord (*n* = 3), while the transcripts of the GDF15 low-affinity receptors, Tgfbr1 and Tgfbr2 (variants 1 and 2) were detected. Moreover, we evidenced that *Tgfbr1* levels were increased in the *SOD1^G93A^* spinal cord (Figure 6A). Our results suggest that GDF15-TGFβ-R1 axis in motoneurons could be a therapeutic target for ALS. Therefore, we analyzed the expression profile in the spinal cords of wildtype and *SOD1^G93A^* mice at 3 months of age. TGFβ-R1 was found to be expressed in nearly all ChAT-positive motoneurons in the spinal cords of both wildtype and *SOD1^G93A^* mice (Figure 6B). In addition, we detected TGFβ-R1-expressing cells in the white matter that had the morphology of radial glia (not shown). Of note, TGFβ-R1 was also observed in glial cells reminiscent of microglia, consistent with the role of TGF-β1 signalling in microglial cells [53,54].

These results demonstrate expression of GDF15 and HB-EGF receptors and their differential regulation in the spinal cord of ALS mice. This finding further supports a potential role of these factors in the pathophysiology of the peripheral motor system.

In summary, using primary culture of motoneurons, our study revealed the regenerative and death-protective effect of human dental pulp cell secretome under ALS conditions. Furthermore, GDF-15 and HB-EGF present in the secretome protect ALS motoneurons exclusively under oxidative stress conditions which confer disease-specific effects on these factors.

## 4. Discussion

Despite extensive research in both fundamental and clinical fields, ALS remains a disease with no effective treatment. The ongoing research on ALS disease mechanisms is therefore of high importance for discovering new therapies to enhance a patient’s quality of life and substantially prolong their life expectancy. Since the pioneering work of Wang et al. on the ALS mouse model, the therapeutic benefits of the DPSC secretome appear as promising therapeutic means for ALS [29]. These cells are capable of secreting neurotrophic factors that are essential for neuronal survival and neurite growth, which make them promising for use in therapeutic procedures [24,25].

Our in vitro study has clarified to some extent the mechanism of the survival effect of DPSC secretome on ALS *SOD1^G93A^* mice by directly influencing the survival of motoneurons. DPSCs-CM has a differential effect on axon outgrowth depending on the expression of SOD1 mutation, but does not change the synaptically driven electrical activity of wildtype or *SOD1^G93A^* neurons. Moreover, GDF15 and HB-EGF, which we have previously found to be secreted by DPSCs-CM, have a neuroprotective effect on *SOD1^G93A^* motoneurons only after exposure to oxidative stress.

DPSCs-CM is composed of many proteins including trophic factors [39,55,56]. Consistent with the effects on motoneuron survival that we have described here, the DPSCs-CM contains factors that are well known to promote neuronal survival and neurite growth during development and adulthood, such as NT-3 or VEGF [57,58]. Additional studies have also shown the expression of BDNF, CNTF, and GDNF by DPSCs, mainly at the transcriptional levels. Our CM preparation does not show a high representability of BDNF and GDNF, although CNTF was not evaluated in our previous study. It is therefore reasonable to propose that the additional effect of several factors allows an optimal survival of motoneurons, finally at a similar extent as that obtained with the combination of the NTFs we used. In addition to the survival effect, DPSCs-CM has no detrimental effect on the synaptic network, which was an important point to verify as hyperexcitability or hypoexcitability are hallmarks of ALS [59]. By rescuing motoneurons from injury, DPSCs could also preserve their functional role which further renders them suitable for in vivo applications.

The beneficial effect of the CM on the survival of wildtype motoneurons seems to be specific to DPSCs, as the CM from ASCs or skin fibroblasts did not promote motoneuron survival, as also observed with fibroblasts on trigeminal motoneurons [3]. Concerning ASCs, their secretome contains various growth and neurotrophic factors that were shown to confer neuroprotective benefits when administrated in SOD1 mutant mice [60,61]. This protective effect of ASCs in a mouse model of ALS is corroborated with their ability to rescue motoneurons from the neurotoxicity of astrocytes derived from ALS patients [62].

While DPSCs-CM promotes the survival of both wildtype and *SOD1^G93A^* motoneuron, our work reveals that DPSCs-CM specifically induces axon outgrowth in wildtype motoneurons but not in *SOD1^G93A^* motoneurons. This intriguing observation parallels recent work on early axonal transport defects, which are observed well before the onset of symptoms in ALS mice [63]. They show that BDNF is able to stimulate the anterograde transport of signaling endosomes in wildtype embryonic motoneurons, but not those expressing *SOD1^G93A^*, due to increased expression of the truncated kinase-deficient form of TrkB and p75^NTR^ at the muscle, sciatic nerve, and Schwann cell levels [63]. Here, our study identified SOD1^G93A^-associated cell-autonomous signaling defects in axonal growth that may involve, as suggested by the previous study [63], differential expression of receptors or their signaling components of the NTFs secreted by DPSCs.

As mentioned above, the DPSC secretome is a complex medium composed of a plethora of soluble factors [40]. Among them, we identified two candidates, GDF15 and HB-EGF, we thought would be of interest, based on the following considerations. Both proteins were shown to confer neuroprotection [50,51,64,65], they have not been studied in the context of ALS, and they were not detected in the secretome of ASCs [66]. We showed that GDF15 or HB-EGF do not increase the survival of wildtype or *SOD1^G93A^*-expressing motoneurons. While the effect of HB-EGF on motoneuron survival has not been documented to our knowledge, HB-EGF has been reported to confer neuroprotection following ischemic injury [50,67]. This suggests that HB-EGF does not have an effect on naturally occurring developmental death but is more associated with degenerative processes in the adult as shown in an experimental model of Alzheimer’s disease [68]. We previously demonstrated that only motoneurons that express ALS-linked mutant forms of SOD1 have increased susceptibility to NO, while the response to excitotoxicity or trophic factor deprivation is comparable to that of the wild type [44]. We found that exogenous HB-EGF can rescue SOD1^G93A^-expressing motoneurons from NO-induced death, supporting the proposition that HB-EGF is a promising neuroprotective factor for adult-onset neurodegenerative disorders including AD and ALS.

Contrary to our results, GDF15 has been shown to contribute to neuronal survival, including in spinal motoneurons [51]. The different culture conditions, age of embryos (E12.5 vs. E13.5), and the isolation of motoneurons from the lumbar or whole spinal cord [51,69] might explain this discrepancy. However, it is worth noting that *gdf15*-deficient mice display a 19% loss of trigeminal and facial motoneurons, and 21% loss of lumbar motoneurons [51]. The discrepancy between our study performed on motoneurons purified from the whole spinal cord and the one carried out on motoneurons from the lumbar part of the spinal cord [51] suggests that GDF15 may indeed be beneficial for some motoneuron subpopulations, as already observed for hepatocyte growth factor [70].

Interestingly, while GDF15 and HB-EGF do not affect spinal motoneurons under conditions of NTF deprivation, they are able to protect *SOD1^G93A^* motoneurons from NO-induced death. This suggests a protective effect under pathological conditions. In a model of spinal cord injury, GDF15 was found to inhibit the oxidative-stress-dependent ferroptotic death of neurons, and RNA-interference-mediated silencing of GDF15 reduced the locomotor recovery of mice [71]. In a mouse model of spinal muscular atrophy (SMA), a longitudinal transcriptomic study of vulnerable and resistant motoneuron pools showed that in SMA-resistant ocular motoneurons, *gdf15* was highly upregulated [72]. In addition, GDF15 was able to rescue human motoneurons derived from induced pluripotent stem cells, IPSCs, from degeneration. Jennings et al. recently reported an elevation of GDF15 in the serum of patients with Charcot–Marie–Tooth disease and a mouse model of this disorder, proposing that the induction of GDF15 is an adaptive response to stress to promote peripheral neuron regeneration [73].

To complete our study, we conducted a quantitative transcriptomic analysis of the receptors for HB-EGF, *Egfr* and *ErbB4*, and GDF15, *Gfral*, *Tgfbr1*, and *Tgfbr2* in the lumbar spinal cords of wildtype and *SOD1^G93A^* mice. Transcripts of both HB-EGF receptors were expressed in the spinal cord. Consistent with a previous study [74], we observed down-regulation of *ErbB4* in the spinal cords of *SOD1^G93A^* mice. ErbB4 is highly expressed at the membrane of motoneurons and at the time of disease onset, ErbB4 immunoreactivity is reduced in some type of motoneurons, followed by greater loss of ErbB4 in the remaining *SOD1^G93A^* motoneurons at symptomatic stages [74]. Of note, a decreased ErbB4 immunoreactivity was also observed in motoneurons of patients with sporadic ALS and was associated with cytoplasmic aggregation of TDP-43 [75]. Therefore, strategies aimed to increase ErB4 receptor or related signaling pathways could have neuroprotective effects in ALS, as has been previously shown with the HGF receptor MET [76].

Transcripts of GFRAL, the high-affinity receptor of GDF15, were not detected in the spinal cord, which is consistent with expression exclusively in the hindbrain [77]. However, GDF15 can bind to its low-affinity receptor, TGFβ-R. TGFβ-R signaling has been well-studied as its ligand TGFβ plays significant role under inflammatory conditions, as observed in ALS [78]. Notably, microglial activation is a hallmark of ALS and therefore it was not surprising that TGFβ-R1 expression reveals the presence of microglia specifically in the SOD1^G93A^ spinal cord, which could explain the increase in its transcript expression. Interestingly, we also observed TGFβ-R1 expression in the soma of motoneurons, whose activation could protect them from oxidative stress as we demonstrated *in vitro*. Overall, these results indicate that GDF15 signaling by TGFβ-R1 may be a novel therapeutic strategy for the treatment of ALS at an early stage of the disease.

Our findings have important clinical implications for improving spinal motoneuron survival and in particular under oxidative stress, a condition encountered during the neurodegenerative process of ALS. Moreover, our previous work demonstrated the regenerative properties of DPSCs on sensory neurons [39] which are also affected during ALS progression in humans [79]. Altogether, these results should encourage continued development of therapeutic strategies. However, the effects of these factors on other neuronal structures that are also affected in ALS need also to be addressed.

Our study also has some limitations as it underestimates the properties of DPSCs-CM when considering the non-cell-autonomous components of motoneuron degeneration in ALS. The regenerative effects of DPSCs in ALS mice could indeed also be mediated by other cell types including glial cells, as shown with ASCs [62]. It would therefore be interesting to evaluate the neuroprotective effects of DPSCs-CM, GDF15, and HB-EGF on human motoneurons challenged with astrocytes, oligodendrocytes, or microglia derived from ALS patient IPSCs. Moreover, GDF15 and HB-EGF act on many other cell types which necessitates development of therapeutic strategies that specifically target the spinal cord or the motoneurons.

## 5. Conclusions

In conclusion, our study reveals the therapeutic potential of DPSC secretome in promoting motoneuron survival and also provides a first step in understanding the key components responsible for this therapeutic effect. The main areas of interest for future studies should include focus on other factors that have never been explored, to rescue motoneurons from death as well to assess new combinations of complementary molecules. We believe that such discoveries will drive the development of new strategies for specifically delivering active and cell-state-dependent cocktails of secretome-derived molecules for the treatment of ALS.

## Figures and Tables

**Figure 1 biomedicines-11-02152-f001:**
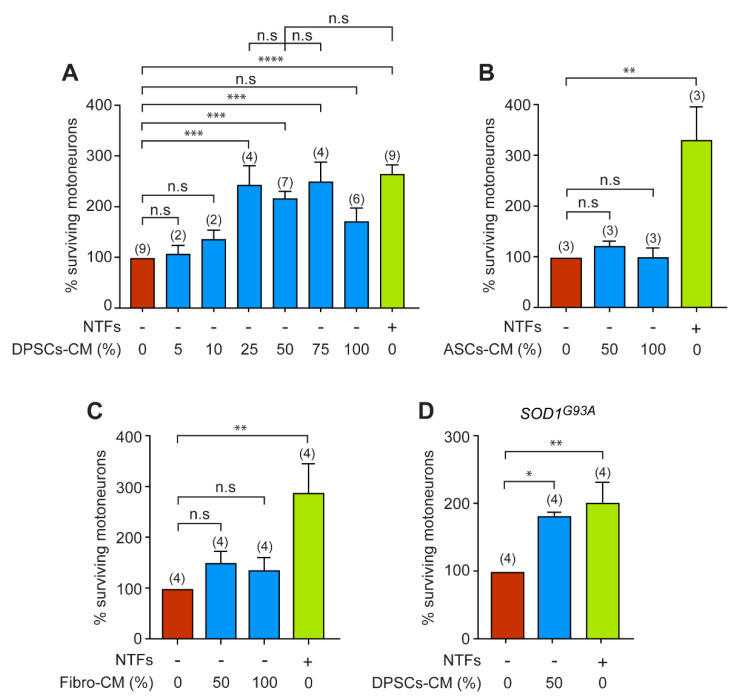
DPSCs-CM increases the survival of wildtype and *SOD1^G93A^* motoneurons.the number of independent experiments (each conducted in triplicate or quadruplicate) is indicated in brackets. (**A**) Motoneurons were immunopurified from the spinal cord of E12.5 mouse embryos and cultured in the absence (-) of a cocktail of NTFs and with increasing concentration of DPSCs-CM. Motoneurons were also cultured in the presence of NTFs (+) only. The percentage of surviving motoneurons, expressed relative to the condition without any trophic support, was calculated after 24 h. Data are means ± SEM and the number of independent experiments (each conducted in triplicate or quadruplicate) is indicated in brackets. Note that for the low concentrations of DPSCs-CM (5% and 10%), which are not relevant for motoneuron survival, the experiments were repeated twice independently, each time in triplicate. One-way ANOVA followed by Tukey’s post hoc test, * *p* < 0.05, ** *p* < 0.01, *** *p* < 0.001, **** *p* < 0.0001, n.s, non-significant. (**B**,**C**) Motoneurons were isolated and seeded in the absence of NTFs and with the same optimal dose previously described (50%), or only (100%) with ASCs-CM (**B**) or Fibro-CM (**C**). The number of surviving motoneurons was determined 24 h later and expressed relative to the survival in the absence of NTFs. All values are expressed as the means ± SEM of three (**B**) or four (**C**) independent experiments (triplicate or quadruplicate). One-way ANOVA followed by Tukey’s test, ** *p* < 0.01, n.s, non-significant. (**D**) Mutant *SOD1^G93A^* motoneurons were cultured with DPSCs-CM (at 50%) or NTFs. Motoneuron survival was determined 24 h later and expressed relative to the basal condition (without NTFs). Data represent the mean ± SEM of triplicates or quadruplicates of four independent experiments.

**Figure 2 biomedicines-11-02152-f002:**
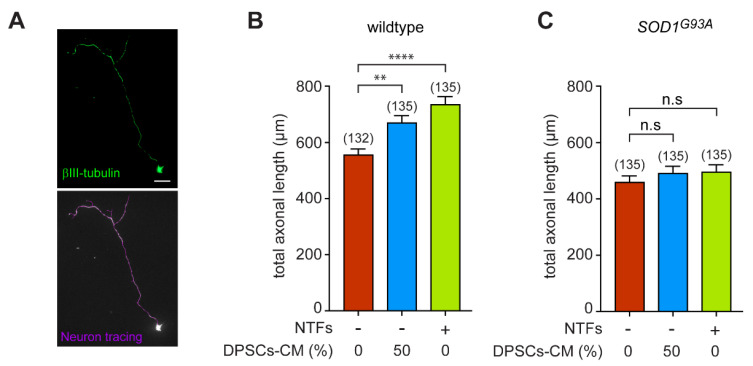
DPSCs-CM induces axon outgrowth of wildtype but not *SOD1^G93A^* motoneurons. The number in brackets indicating the total number of motoneurons measured. (**A**) Representative immunostaining of a wildtype motoneuron with anti-βIII-tubulin antibody. The total axon length that also included collaterals of motoneurons (violet trace) was measured with ImageJ using the NeuronJ plugin. Scale bar = 20 µm. (**B**,**C**) Freshly purified motoneurons from E12.5 wildtype (**B**) or SOD1^G93A^ (**C**) embryos were treated (or not) with DPSCs-CM at the optimal concentration of 50% and NTFs. Measurement of the total axon length was performed after 24 h of culture and values are expressed relative to the basal condition (in the absence of NTFs). Graphs represent the mean value ± SEM of three independent experiments, the number in brackets indicating the total number of motoneurons measured. Data were analyzed by one-way ANOVA followed by Tukey’s post hoc test. ** *p* < 0.01, **** *p* < 0.0001, and n.s, non-significant.

**Figure 3 biomedicines-11-02152-f003:**
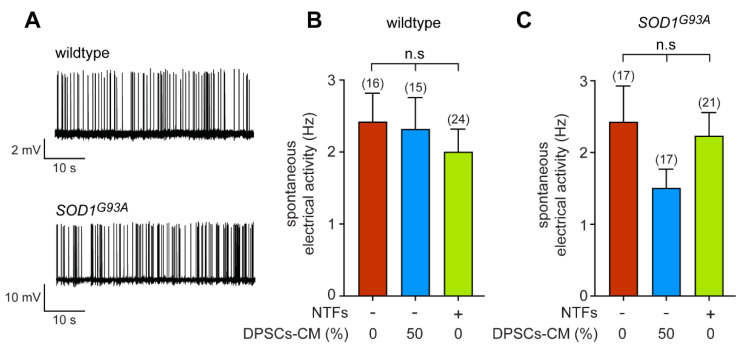
DPSCs-CM does not modify the spontaneous electrical activity of motoneurons. (**A**) Representative trace of recordings of spontaneous electrical activity of wildtype and *SOD1^G93A^* neurons after 7 DIV in the presence of NTFs. (**B**,**C**) Motoneuron-enriched cultures were prepared from *Hb9::GFP* wildtype (**B**) and SOD1mutant mice (**C**) and cultured for 7 DIV in the absence of NTFs, with 50% DPSCs-CM or in the presence of trophic support. Spontaneous electrical activity was measured using the loose-patch technique and spike frequency calculated. The total number of recorded cells is indicated in brackets. Values are means ± SEM of three and four independent experiments for wildtype and *SOD1^G93A^* conditions, respectively. Data were analyzed by one-way ANOVA followed by Tukey’s post hoc test, n.s, non-significant.

**Figure 4 biomedicines-11-02152-f004:**
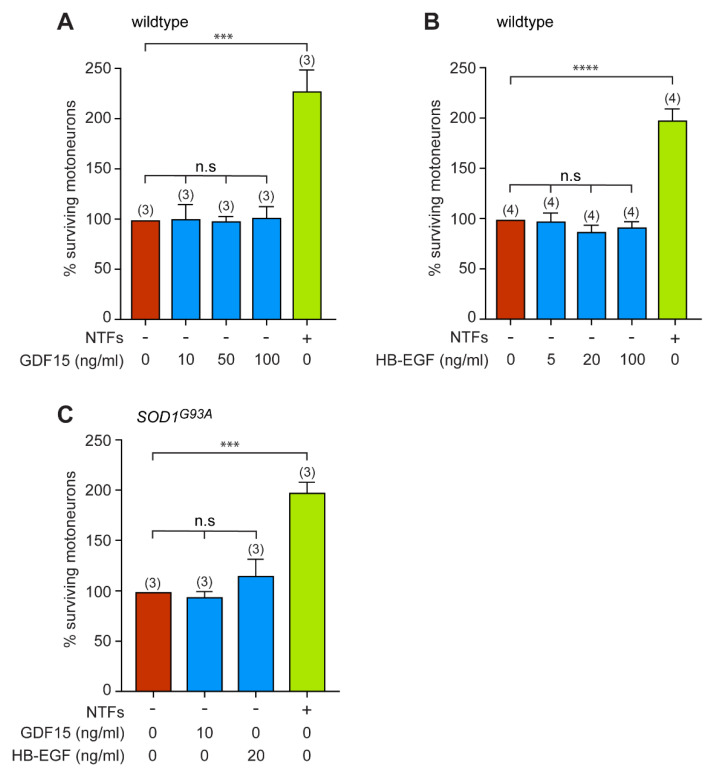
Recombinant GDF15 and HB-EGF do not provide trophic support to either wildtype or SOD1 mutant motoneurons. (**A**) Motoneurons were plated in basal conditions (without NTFs) and treated with indicated concentrations of recombinant GDF15 (or cultured with NTFs only). The survival of motoneurons was determined after 24 h. (**B**) Mouse motoneurons were treated with increasing concentrations (5, 20, and 100 ng/mL) of HB-EGF at the time of seeding. Twenty-four hours later, motoneuron survival was assessed. (**C**) Motoneurons immunopurified from *SOD1^G93A^* E12.5 embryos were incubated in the absence of NTFs with either GDF15 (10 ng/mL) or HB-EGF (20 ng/mL). The percentage of surviving *SOD1^G93A^* motoneurons was determined after 24h of treatment. Data represent the mean values ± SEM of triplicates of three (**A**,**C**) and four(**B**) independent experiments (number in brackets). Data were analyzed by one-way ANOVA followed by Tukey’s post hoc test, *** *p* < 0.001, **** *p* < 0.0001, n.s, non-significant.

**Figure 5 biomedicines-11-02152-f005:**
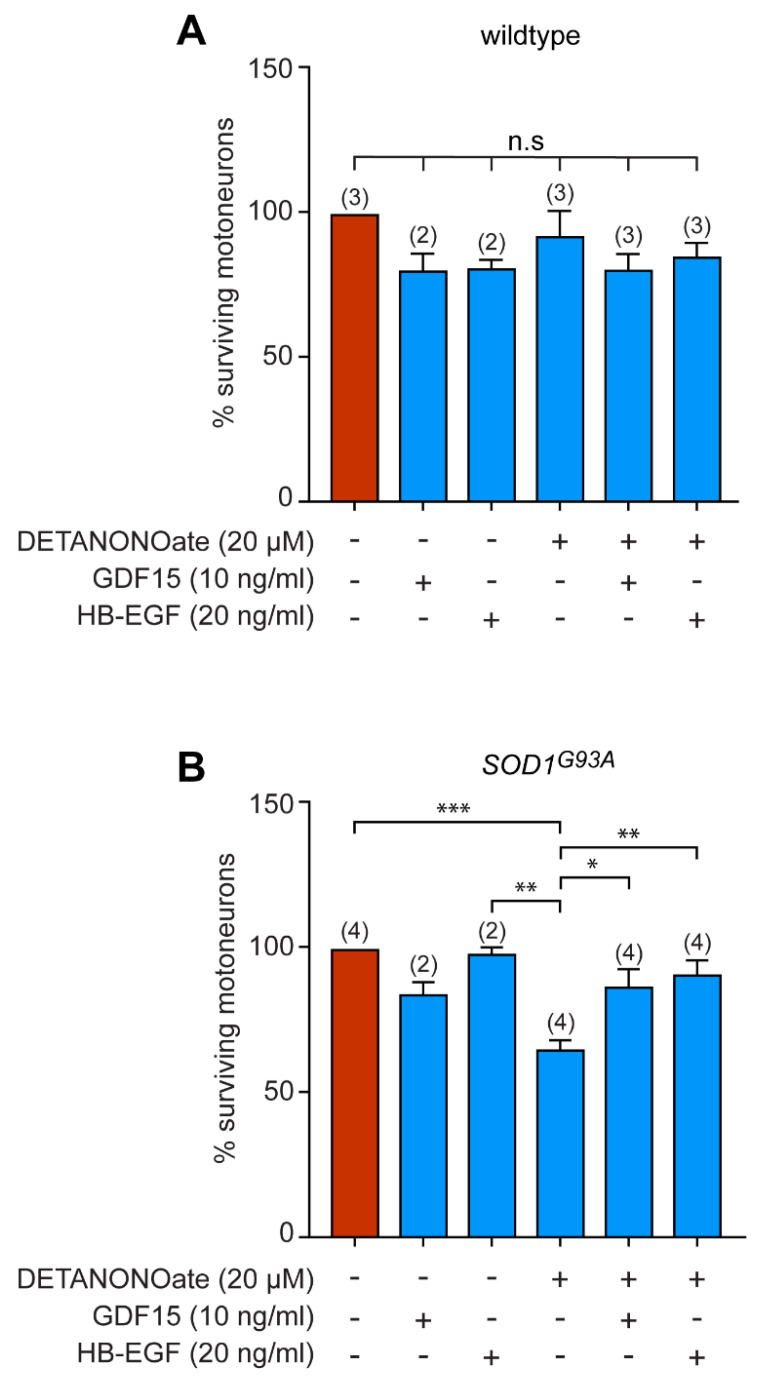
GDF15 and HB-EGF protect *SOD1^G93A^* motoneurons from NO-induced death. the number of independent experiments (each performed in triplicate or quadruplicate) is indicated in brackets. (**A**) Wildtype motoneurons were cultured for 24 h in the presence of NTFs and incubated with the NO donor DETANONOate (20 µM) in combination with GDF15 (10 ng/mL) and HB-EGF (20 ng/mL). The percentage of surviving motoneurons was determined 48 h later. (**B**) SOD1^G93A^-expressing motoneurons were maintained in culture for 24 h and treated (or not) with DETANONOate (20 µM), GDF15 (10 ng/mL), or HB-EGF (20 ng/mL) for 48 h. The number of surviving motoneurons is expressed as a percentage of the number of motoneurons in the control condition (in the presence of NTFs only). Histograms show mean values ± SEM; the number of independent experiments (each performed in triplicate or quadruplicate) is indicated in brackets. One-way ANOVA followed by Tukey’s post hoc test, * *p* < 0.05, ** *p* < 0.01, *** *p* < 0.001.

**Figure 6 biomedicines-11-02152-f006:**
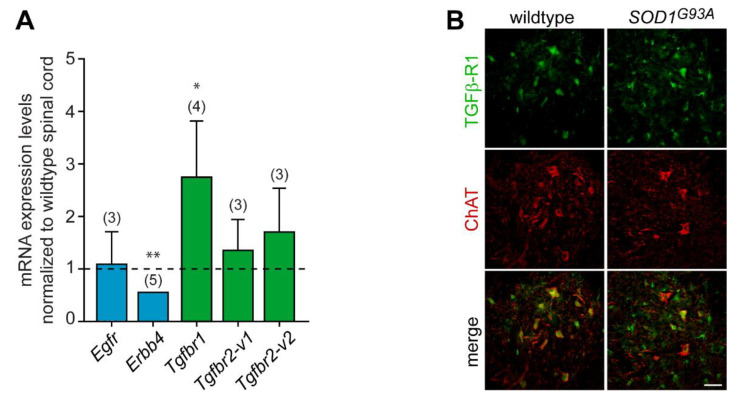
GDF15 and HB-EGF receptors are differentially expressed in the spinal cords of presymptomatic mice. (**A**) Quantitative RT-PCR was performed on total RNA isolated from the spinal cords of 3-month-old wildtype and SOD1^G93A^ mice. Transcript levels of *Egfr*, *Erbb4*, *Gfral*, *Tgfbr1*, and *Tgfbr2* were expressed relative to polymerase (RNA) II polypeptide J (*Polr2J*) transcript. The expression levels of each transcript in the spinal cords of *SOD1^G93A^* mice were normalized to those obtained in the spinal cords of wildtype mice (represented by the dashed line). Values are means ± SEM, the number of mice is indicated in brackets. Data were analyzed by Mann–Whitney test. * *p* < 0.05, ** *p* < 0.01 vs. wildtype spinal cord. (**B**) Lumbar spinal cord sections of wildtype and SOD1^G93A^ mice at 3 months of age were immunostained with antibodies against TGFβ-R1 (in green) and ChAT (in red). Scale bar, 50 µm.

## Data Availability

The data presented in this study are available on request from the corresponding author.

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
