# Peer review of "The Secretome of Human Dental Pulp Stem Cells and Its Components GDF15 and HB-EGF Protect Amyotrophic Lateral Sclerosis Motoneurons against Death"

_biomedicines, 2023, doi:10.3390/biomedicines11082152_

Round 1
Reviewer 1 Report
The manuscript by Younes R et al n evaluated the effects of dental pulp stem cells (DPSCs) secretome on the survival, axon outgrowth, synaptically-driven activity of mouse primary motoneurons from wildtype and ALS model mice. They shown that the DPSCs secretome promotes axon outgrowth of wildtype, but not ALS model motor neurons. Moreover, the author demonstrated that GDF and HB-EGF, which are secreted by DPSCs, protect motor neurons form ALS model mice against nitric oxide-induced death. From the results obtained in these results, the authors concluded that the secretome of DPSCs may be a novel therapeutic strategy for the treatment of ALS at an early stage of the disease. But, I feel that this conclusion over interpret the evidence. I have several concerns with their experiments and results. In my opinion, this manuscript is not recommended for publication in its present form, but may accept as the paper after major revision.
1 More Data
The authors explained GDF and HB-EGF are secreted by DPSCs in this manuscript, but the authors should give some references and convincing evidence about this point. Especially, the author should show the data about the concentrations of GDF and HB-EGF secreted by DPSCs in this condition. I recommend you to add more data about this point.
2 the numbers of experiment
the authors should add the numbers of experiment in Fig. 1 4, 5 and 6. More importantly, the authors should indicate the reason that the number (n=2) is enough in the experiment in Fig 1.
3 Abstract
Abbreviations used should be defined once the first time they appear in the text.
Abbreviations used should be defined once the first time they appear in the text.
Reviewer 2 Report
Younes and colleagues in the present article entitled ‘The secretome of human dental pulp stem cells, GDF15 and HB-EGF confer neuroprotective effects to spinal motoneurons of ALS mice’, present the findings of an in vitro study that investigates the effects of dental pulp stem cell secretome (DPSCs-CM) on the survival and axon growth of wildtype and SOD1G93A motoneurons, as well as its impact on synaptically-driven electrical activity. The study identifies two proteins, GDF15 and HB-EGF, within the DPSCs-CM that exhibit a neuroprotective effect on SOD1G93A motoneurons specifically under oxidative stress conditions. The authors highlight the presence of various prosurvival factors in DPSCs-CM, such as NT-3 and VEGF, and discuss their potential contribution to motoneuron survival and neurite growth. The study further compares the effects of DPSCs-CM with secretomes from other cell types and emphasizes the specific beneficial impact of DPSCs-CM on wildtype motoneurons. Additionally, the authors suggest that the differential response of wildtype and SOD1G93A motoneurons to DPSCs-CM may be associated with signaling defects in axonal growth, potentially involving receptors or signaling components of the neurotrophic factors secreted by DPSCs. They highlight the neuroprotective properties of GDF15 and HB-EGF under pathological conditions and discuss their potential therapeutic relevance for neurodegenerative disorders like ALS. The manuscript concludes by highlighting the importance of further research to explore the effects of DPSCs-CM, GDF15, and HB-EGF on human motoneurons challenged with different cell types derived from ALS patient cells.
In general, I think the idea of this article is really interesting and the authors’ fascinating observations on this timely topic may be of interest to the readers of Biomedicines. However, some comments, as well as some crucial evidence that should be included to support the author’s argumentation, needed to be addressed to improve the quality of the manuscript, its adequacy, and its readability prior to the publication in the present form. My overall judgment is to publish this paper after the authors have carefully considered my suggestions below, in particular reshaping parts of the ‘Introduction’ and ‘Methods’ sections by adding more evidence.
Please consider the following comments:
• I suggest changing the title. In my opinion, in the present form it is too wordy and it seems to be not enough clear and specific.
• Abstract: In my opinion, Authors should consider rephrasing this section. According to the Journal’s guidelines, the Abstract should contain most of the following kinds of information in brief form. Please, consider giving a more synthetic overview of the paper's key points: I would suggest rephrasing the results and conclusion to make them clear for readers to understand.
• A graphical abstract that will visually summarize the main findings of the manuscript is highly recommended.
• Introduction: The introduction provides a good overview of amyotrophic lateral sclerosis (ALS) as a neurodegenerative disease. However, it lacks specific details regarding the current state of research and recent advancements in the field: in this regard, I believe that more information about pathophysiology and core features of the diseases being studied (diabetes, multiple sclerosis, migraine) and their potential associations with Alzheimer's disease (AD) will provide a better and more accurate background, because as it stands, this information is not highlighted in the text. In this regard, I would suggest to add more information on pathological neural substrates of neurodegeneration in AD, specifically on structural as well as functional abnormalities of specific brain regions (i.e., hippocampus and prefrontal cortex), and on related and on related effects on patients’ cognitive impairments (DOI: 10.3390/biomedicines11030945). In my opinion, authors could further explore significant structural brain alterations and impaired brain circuits in AD (https://doi.org/10.1016/j.neubiorev.2023.105163), and focus on relationship between the molecular regulation of higher-order neural circuits and neuropathological alterations in this neurodegenerative disorder (DOI: 10.3390/biomedicines10122999; https://doi.org/10.3390/biomedicines10092220).
• Introduction: This section briefly mentions neurotrophic factors (NTFs) as potential therapeutic candidates for ALS, but it would benefit from more context and a stronger rationale for exploring dental pulp stem cells (DPSCs) and their secretome.
• Preparation of conditioned media: I would suggest rewriting this section, as the description of the preparation of conditioned media (CM) from DPSCs, adipose-derived stem cells (ASCs), and fibroblasts lacks specific details, such as the exact concentrations of the enzymes used for digestion or the duration of the incubation steps.
• Animals: This section on animals provides basic information about the strains used but does not mention the number of animals used in the study or any efforts made to minimize animal suffering or follow ethical guidelines.
• Results: I suggest rewriting this section more accurately. To properly present experimental findings, I think that authors should provide full statistical details (like degree of freedom or post-hoc utilized), to ensure in-depth understanding and replicability of the findings.
• In my opinion, I think the ‘Conclusions’ paragraph would benefit from some thoughtful as well as in-depth considerations by the authors, because as it stands, it is very descriptive but not enough theoretical as a discussion should be. Authors should make an effort, trying to explain the theoretical implication as well as the translational application of their research.
• References: Authors should consider revising the bibliography, as there are several incorrect citations. Indeed, according to the Journal’s guidelines, they should provide the abbreviated journal name in italics, the year of publication in bold, the volume number in italics for all the references.
• Overall, the writing style could be improved by ensuring clarity and coherence, using appropriate scientific terminology, and organizing the information in a logical flow.
I hope that, after these careful revisions, this paper can meet the Journal’s high standards for publication.
I am available for a new round of revision of this article.
I declare no conflict of interest regarding this manuscript.
Best regards,
Reviewer
Minor editing of English language required.
Round 2
Reviewer 1 Report
The study appears to be of interest, whereas the experiments have some problems. In my opinion, this manuscript is not recommended for publication in its present form.
I pointed out the following a point out the number of samples in fig.1 . the authors need to describe the reason that the number of samples (n=2) is enough in the experiment in Fig 1.
You may be confused, I am not criticizing the range of concentrations (2 point) considered in this manuscript.
Author Response
Please see the attachment : Reply to the Reviewer 1

Reviewer 2 Report
Younes and colleagues in the present article entitled ‘The secretome of human dental pulp stem cells, GDF15 and HB-EGF confer neuroprotective effects to spinal motoneurons of ALS mice’, presented the findings of an in vitro study that investigates the effects of dental pulp stem cell secretome (DPSCs-CM) on the survival and axon growth of wildtype and SOD1G93A motoneurons, as well as its impact on synaptically-driven electrical activity.
I only have one last suggestion to do: to further enhance the understanding of these results, it would be beneficial to explore several aspects in more depth. This study focuses on the understanding of the Amyotrophic lateral sclerosis (ALS) disease and current therapeutic challenges; still, I believe that it would be beneficial to include more information about the neural substrates involved in ALS pathogenesis. Neural substrates refer to the specific brain regions, neural circuits, and cellular mechanisms that underlie a particular disease or condition. In the case of ALS, understanding the neural substrates involved can provide valuable insights into the complex molecular pathways and cell types implicated in disease progression. Therefore, it is important to highlight the relevance of neural substrates when discussing potential therapeutic strategies (DOI: 10.3390/cells11162607; https://doi.org/10.1016/j.neubiorev.2023.105163; DOI: 10.17219/acem/165944; https://doi.org/10.3389/fpsyt.2023.1225755). By deepening this concept, the Authors here can highlight the importance of considering neural substrates in the context of ALS research and treatment development.
Overall, this is a timely and needed work, and I only had one last comment to do, to fully support the authors’ claims. My overall judgment is this article but only after the authors have carefully considered my last suggestion.
I am always available for other reviews of such interesting and important articles.
Happy to help.
Reviewer
Author Response
Please see the attachment : Reply to the Reviewer 2

Round 3
Reviewer 1 Report
I recommend this article accept in present form.
Author Response
Thank you for accepting our article for publication
Reviewer 2 Report
The authors did an excellent job clarifying all the questions I have raised in my previous round of review. Currently, this paper is a well-written, timely piece of research that presented the findings of an in vitro study that investigates the effects of dental pulp stem cell secretome (DPSCs-CM) on the survival and axon growth of wildtype and SOD1G93A motoneurons, as well as its impact on synaptically-driven electrical activity.
Overall, this is a timely and needed work. It is well researched and nicely written, with a good balance between descriptive and narrative text.
I believe that this paper does not need a further revision, therefore the manuscript meets the Journal’s high standards for publication.
I am always available for other reviews of such interesting and important articles.
Reviewer
Author Response

(The authors gave the same response as above.)
